# Impact of Early SARS-CoV-2 Antiviral Therapy on Disease Progression

**DOI:** 10.3390/v15010071

**Published:** 2022-12-27

**Authors:** Andrea De Vito, Agnese Colpani, Laura Saderi, Mariangela Puci, Beatrice Zauli, Vito Fiore, Marco Fois, Maria Chiara Meloni, Alessandra Bitti, Cosimo Di Castri, Ivana Maida, Sergio Babudieri, Giovanni Sotgiu, Giordano Madeddu

**Affiliations:** 1Unit of Infectious Disease, Department of Medicine, Surgery and Pharmacy, University of Sassari, 07100 Sassari, Italy; 2Clinical Epidemiology and Medical Statistics Unit, Department of Medicine, Surgery and Pharmacy, University of Sassari, 07100 Sassari, Italy

**Keywords:** COVID-19, SARS-CoV-2, vaccination, disease progression, molnupiravir, nirmatrelvir/ritonavir, remdesivir, antiviral treatment, monoclonal antibody

## Abstract

Since the start of the SARS-CoV-2 pandemic, several treatments have been proposed to prevent the progression of the disease. Currently, three antiviral (molnupiravir, nirmaltrevir/r, remdesivir) and two monoclonal antibodies (casirivimab/imdevimab and sotrovimab) are available in Italy. Therefore, we aimed to evaluate the presence of risk factors associated with disease progression. We conducted a retrospective cohort study, including all patients with a confirmed diagnosis of SARS-CoV-2 evaluated between 01/01/2022 ad 10/05/2022 by our Unit of Infectious Diseases in Sassari. We defined disease progression as the necessity of starting O2 therapy. According to AIFA (Italian Medicines Agency) indications, preventive treatment was prescribed in patients with recent symptoms onset (≤five days), no need for oxygen supplementation, and risk factors for disease progression. Subgroup differences in quantitative variables were evaluated using Student’s *t*-test. Pearson chi-square or Fisher’s exact tests were used to assess differences for qualitative variables. Multivariate logistic regression modelling was performed to determine factors associated with progression. A two-tailed p-value less than 0.05 was considered statistically significant. All statistical analyses were performed with STATA version 17 (StataCorp, College Station, TX, USA). We included 1145 people with SARS-CoV-2 diagnosis, of which 336 (29.3%) developed severe disease with oxygen supplementation. In multivariate logistic regression analysis, age, dementia, haematologic tumors, heart failure, dyspnoea or fever at first evaluation, having ground glass opacities or consolidation at the first CT scan, and bacteria coinfection were associated with an increased risk of disease progression. Vaccination (at least two doses) and early treatment with antiviral or monoclonal antibodies were associated with a lower risk of disease progression. In conclusion, our study showed that vaccination and early treatment with antiviral and/or monoclonal antibodies significantly reduce the risk of disease progression.

## 1. Introduction

At the end of December 2019, several cases of an unknown respiratory syndrome were reported in Wuhan (Hubei Province, China). On 9 January 2020, Severe Acute Respiratory Syndrome – Coronavirus 2 (SARS-CoV2), the causative agent of the novel Chinese respiratory syndrome, was finally identified [1,2,3].

The estimated cumulative incidence of SARS-CoV-2 infection cases is half a billion globally, with more than six million deaths caused by COVID-19. To date, 23 million cases with more than 178 thousand deaths have been recorded in Italy [4].

The most prevalent symptoms of COVID-19 are fever, cough, and dyspnoea; a low proportion complain of gastrointestinal symptoms, anosmia, dysgeusia, headache, and skin lesions [5,6,7,8,9,10]. However, the disease can progress to life-threatening systemic inflammation, respiratory failure, and multiorgan dysfunction [11,12]. 

At the end of 2020, mRNA vaccines were distributed worldwide and showed high efficacy and safety in disease prevention [13,14,15,16]. 

Concomitantly, several therapies are available for hospitalised patients with moderate or severe COVID-19 [17,18,19,20]. Three early SARS-CoV-2 antiviral therapies are currently available in Italy. Molnupiravir and remdesivir are inhibitors of SARS-CoV-2 polymerases, while nirmatrelvir/ritonavir(r) inhibits the main protease (M_pro_) [21,22,23]. To date, real-life data about the use and efficacy of these treatments are limited in the literature [24,25]. 

The aim of the present study was to evaluate the effectiveness of antivirals and monoclonal antibodies in preventing disease progression.

## 2. Materials and Methods

An observational, retrospective study was conducted at an Italian university hospital (University of Sassari, Sassari) between 1 January and 31 July 2022. Inclusion criteria were: (i) age ≥ 18 years; (ii) diagnosis of SARS-CoV-2 infection by Polymerase Chain Reaction (PCR) or third-generation antigenic tests; (iii) exclusion of respiratory failure (PaO2/FiO2 > 300). All people included in the study were evaluated in the emergency room, or in wards for those who had a hospital infection.

We collected information on medical history, vaccination status, computer tomography (CT) findings, symptoms at admission, and therapies. 

### 2.1. Treatment Prescription

Antiviral treatments (molnupiravir, nirmatrelvir/ritonavir, and remdesivir) and monoclonal antibodies (casirivimab/imdevimab and sotrovimab) were prescribed in accordance with the recommendations of the Italian Drug Agency. Criteria for their prescription were: recent symptom onset (≤5 days for molnupiravir and nirmatrelvir/ritonavir, ≤7 days for remdesivir and monoclonal antibodies), no need for oxygen supplementation, and high risk of disease progression for the presence of at least one of the following chronic diseases: (i) obesity (body mass index > 30); (ii) diabetes mellitus with organ damage or hemoglobin A1c > 7.5%; (iii) kidney failure; (iv) severe lung disease; (v) severe cardiovascular disease; (vi) immune deficiency; (vii) cancer. Contraindications included (i) estimated glomerular filtration rate (eGFR) <30 ml/min/1.73 m^2^ (only for nirmatrelvir/ritonavir and remdesivir); (ii) pregnancy; (iii) advanced chronic liver disease. In addition, males had to accept the use of condoms for at least three months if their partner was fertile, while fertile females had to accept the use of condoms for at least four days from the end of treatment. 

Regarding treatment choice, when more than one was eligible, the choice was at the discretion of the attending physician. The choice was driven by drug–drug interactions (DDI), severe renal impairment, the necessity of hospital admission, the ability to swallow, and patient frailty.

### 2.2. Outcomes

The effectiveness and safety of antiviral treatments and monoclonal antibodies to decrease the risk of disease progression (i.e., administration of oxygen, non-invasive ventilation, and death) were evaluated. 

Furthermore, predictors of disease progression were investigated.

### 2.3. Statistical Analysis

Quantitative variables were summarised with medians and 25th−75th percentiles (interquartile range), whereas qualitative ones with absolute and relative (percentages) frequencies. The Shapiro–Wilk test was used to assess the normality of quantitative data. The Mann–Whitney test evaluated subgroup differences for quantitative variables. Pearson’s χ^2^ or Fisher’s exact tests were used to assess differences for qualitative variables. Logistic regression analysis was performed to identify factors associated with disease progression. A two-tailed *p*-value less than 0.05 was considered statistically significant. All statistical analyses were performed with STATA version 17 (StatsCorp, TX, USA).

## 3. Results

Between January 1 and July 31, 1581 people were evaluated. Of these, 220 (13.9%) were not enrolled because of missing data. Of the remainder, 216/1361 (15.9%) had moderate/severe acute respiratory distress syndrome (ARDS) with a PaO2/FiO2 < 300 (Figure 1).

Overall, 1145 people were enrolled in the study. The median (IQR) age was 74 (62–83) years, with 417 (36.4%) people > 80 years; 336 (29.3%) developed severe disease needing oxygen supplementation (Table 1).

A total of 389 (34.0%) patients received antiviral therapies; specifically, 242 (21.1%) received molnupiravir, 39 (3.4%) nirmatrelvir/ritonavir, and 108 (9.4%) remdesivir.

People treated with nirmatrelvir/ritonavir were younger and showed a lower comorbidity burden (Table 2). More than half of people treated with remdesivir acquired SARS-CoV-2 infection during hospitalisation. The percentage of disease progression in people treated with remdesivir was twofold higher, without significant differences.

At the logistic regression analysis, an age of >50 years was associated with a higher risk of disease progression, especially for people aged 70–79 (Table 3). Dementia, haematologic cancer, and heart failure were associated with a higher risk of disease progression; the same risk existed in cases of fever or dyspnoea, ground-glass opacities or consolidation at the CT scan, and bacterial coinfections.

By comparison, receiving at least two doses of vaccine and receiving treatment with antiviral or monoclonal antibodies were associated with a lower risk of progression.

## 4. Discussion

Our retrospective study showed that antivirals and monoclonal antibodies were associated with a significantly lower risk of disease progression in COVID-19 patients.

Three clinical trials (MOVe-OUT, EPIC-HR, and PINETREE) demonstrated the efficacy of antivirals [21,23,26]. However, the enrolled population was different when compared with those attending inpatient and outpatient settings: median ages were 42 (MOVeOUT), 45 (EPIC-HR), and 50 (PINETREE) years, whereas in our study it was 74 years. As well, the main comorbidity was obesity, with a low prevalence of other comorbidities. On the other hand, in the PINETREE trial, they enrolled patients with a higher comorbidity burden: patients with diabetes totalled 61.4% (MOVe-OUT), and those with COPD totalled 24% in the PINETREE trial [21,23,26]. 

To the best of our knowledge, this is the first study to describe the efficacy of the three available antiviral drugs in a real-life scenario.

Ronza Najjar-Debbiny et al. enrolled 4,737 treated with nirmatrelvir/ritonavir. They observed that people treated with nirmatrelvir/ritonavir and those with adequate (two or more doses) SARS-CoV-2 vaccination had a lower risk of disease progression [27]. Another study by the same research group evaluated the efficacy of monlupiravir. They observed that people treated with molnupiravir had no differences in composite outcomes (severe disease or death) HR 0.83 (95%CI 0.57–1.21). However, subgroup analyses showed that monlupiravir was associated with a significant decrease in the risk of composite outcomes in older patients, females, and people with inadequate COVID-19 vaccination [28].

Another Israeli study by Ronen Arbel et al. showed that Nirmatrelvir/r reduced the risk of hospitalisation due to COVID-19 only in people older than 64 (HR 0.27 [CI95%] 0.15–0.49), while being male and not having had adequate vaccination was related to a higher risk of hospitalization [22].

Wong et al. published their data about nirmatrelvir/r and monlupiravir use in hospitalised patients in Hong Kong. Both monlupiravir and nirmatrelvir/r showed a beneficial impact on disease progression (HR 0.60 [95%CI 0.52–0.69] and HR 0.57 [95%CI 0.45–0.72], respectively) [25]. The same authors published another study on non-hospitalised people treated with monlupiravir or nirmatrelvir/r. Both treatments reduced mortality risk (monlupiravir HR 0.76 [95%CI 0.61–0.95] and nirmatrelvir/r HR 0.34 [0.22–0.52]) and in-hospital disease progression. By contrast, only nirmatrelvir/r was linked to reduced hospitalisation. However, a significant limitation of this study was the absence of comorbidities. In addition, the percentage of vaccinated people in the nirmatrelvir/r group was double that in the monlupiravir group [29].

Other studies with small sample sizes have been conducted. Gentile et al. enrolled 257 people, of which 146 were treated with monlupiravir and 111 with nirmatrelvir/r. People treated with monlupiravir were older than people treated with nirmatrelvir/r (median age(IQR) 70 (59–79) vs. 60 (40–67), p < 0.001) and with a higher comorbidities burden. Only four people were hospitalised during the follow-up; three had been treated with monlupiravir and one with nirmatrelvir/r [30].

Finally, Piccicacco et al. published a paper on using early remdesivir in a real-life setting, comparing it with people treated with sotrovimab and those who had not received any treatments [31]. People treated with remdesivir or sotrovimab had a lower risk of hospitalisation and emergency department admission due to COVID-19 than those without treatments. No differences were noted between people treated with remdesivir and sotrovimab.

Our data are consistent with previous findings, confirming the benefits of early antiviral treatments on disease progression: monlupiravir had an OR of 0.11, followed by nirmatrelvir/r with 0.22 and remdesivir with 0.26. In addition, vaccination was associated with a lower risk of disease progression (OR 0.27). 

It is crucial to highlight that, in all studies involving treatment with monlupiravir, patients receiving this antiviral had a higher mean age and showed a higher comorbidities burden.

Our study has some limitations. Its monocentric and retrospective nature could affect its generalizability. However, we believe it could be generalised to all countries where antiviral and monoclonal antibodies are available. Furthermore, the number of patients treated with nirmatrelvir/r is poor because of a high comorbidity burden and DDI. 

However, the total sample size was large, and the data collection was complete. In addition, we reported data about the efficacy of the remdesivir three-day course, on which few data are available in the published literature.

## 5. Conclusions

Our study showed that early SARS-CoV-2 antiviral therapy reduces the risk of disease progression and, consequently, hospitalisation and death. However, prospective studies with a more heterogeneous patient profile could confirm the evidence on antiviral drugs’ efficacy in a real-life scenario.

## Figures and Tables

**Figure 1 viruses-15-00071-f001:**
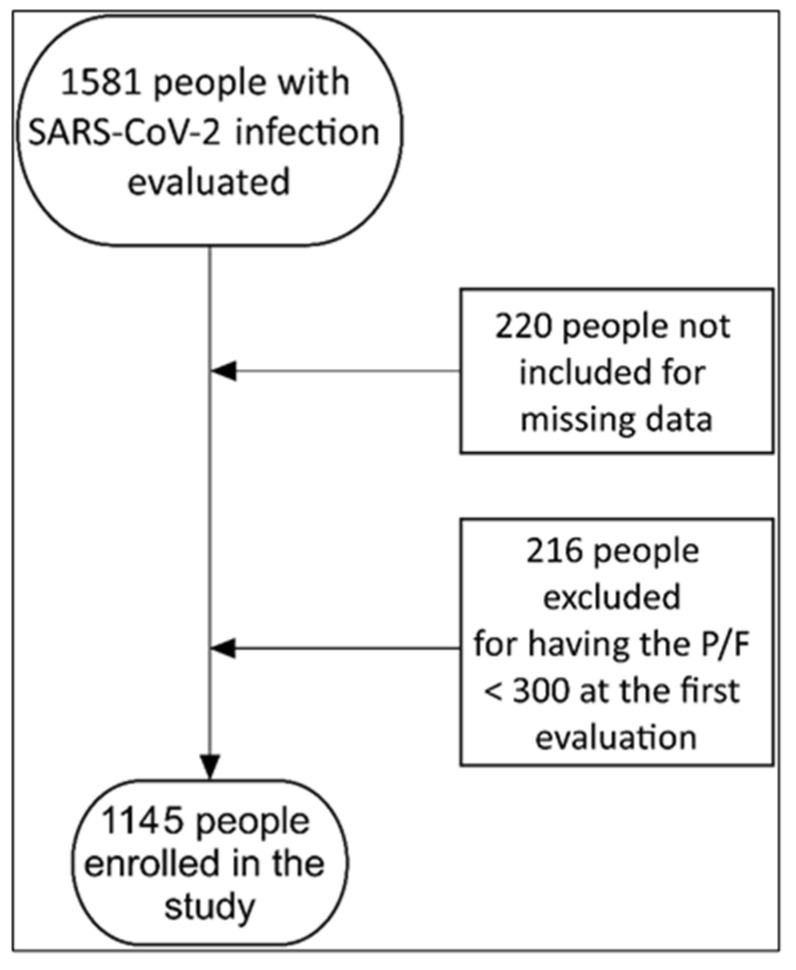
Number of people included in the study and reasons for exclusion.

**Table 1 viruses-15-00071-t001:** Demographic characteristics, comorbidities, symptoms, radiological findings, biochemical features, and treatments of 1145 patients with SARS-CoV-2 infection with or without disease progression.

Variables		Total Cohort (n = 1145)	Non-Severe Disease (n = 809)	Severe Disease (n = 336)	*p*-Value
Males, n (%)		612 (53.5)	426 (52.7)	186 (55.4)	0.40
Median (IQR) age, years		74 (62–83)	72 (59–82)	77.5 (66.5–86.0)	<0.0001
Age group	<50 years	134 (11.7)	118 (1.6)	16 (4.8)	<0.0001
	50–59 years	115 (10.0)	90 (11.1)	25 (7.4)	
	60–69 years	209 (18.3)	147 (18.2)	62 (18.5)	
	70–79 years	270 (23.6)	182 (22.5)	88 (26.2)	
	≥80 years	417 (36.4)	272 (33.6)	145 (43.2)	
**Comorbidities**					
Median (IQR), weight, kg		70 (60–80)	70 (60–80)	70 (62–80)	0.11
BMI > 30 kg/m^2^, n (%)		260 (22.7)	173 (21.4)	87 (25.9)	0.10
Chronic renal failure, n (%)		186 (16.2)	127 (15.7)	59 (17.6)	0.44
Dialysis, n (%)		24 (2.1)	15 (1.9)	9 (2.7)	0.38
Immunodeficit, n (%)		156 (13.6)	104 (12.9)	52 (15.5)	0.24
Transplanted, n (%)		17 (1.5)	10 (1.2)	7 (2.8)	0.28
Rheumatological disease, n (%)		62 (5.4)	42 (5.2)	20 (6.0)	0.60
Decompensated diabetes, n (%)		183 (16.0)	109 (13.5)	74 (22.0)	<0.0001
Diabetes, n (%)		252 (22.0)	172 (21.3)	80 (23.8)	0.34
Chronic liver disease, n (%)		66 (5.8)	42 (5.2)	24 (7.1)	0.20
COPD/Emphysema, n (%)		222 (19.4)	137 (16.9)	85 (25.3)	0.001
Hemoglobinopathies, n (%)		5 (0.49)	4 (0.5)	1 (0.3)	0.65
Neurodevelopmental/neurodegenerative diseases, n (%)		321 (28.0)	203 (25.1)	118 (35.1)	0.001
Dementia, n (%)		176 (15.4)	100 (12.4)	76 (22.6)	<0.0001
Chromosopathies/hypoxia, n (%)		8 (0.7)	4 (0.5)	4 (1.2)	0.20
Neuromuscular disease, n (%)		33 (2.9)	25 (3.1)	8 (2.4)	0.51
Cerebrovascular events, n (%)		134 (11.7)	88 (10.9)	46 (13.7)	0.18
Oncological disease, n (%)		170 (14.9)	133 (16.4)	37 (11.0)	0.02
Metastasis, n (%)		58 (5.1)	41 (5.1)	17 (5.1)	1.00
Terminal cancer, n (%)		20 (1.8)	6 (0.7)	14 (4.2)	<0.0001
Haematological tumours, n (%)		71 (6.2)	44 (5.4)	27 (8.0)	0.10
Haematological tumours in chemotherapy, n (%)		48 (4.2)	30 (3.7)	18 (5.4)	0.21
Cardiovascular diseases, n (%)		417 (36.4)	276 (34.1)	141 (42.0)	0.01
Heart failure, n (%)		370 (32.3)	241 (29.8)	129 (38.4)	0.005
Previous myocardial infarction, n (%)		147 (12.8)	101 (12.5)	46 (13.7)	0.58
Hypertension, n (%)		547 (47.8)	361 (44.6)	186 (55.4)	0.001
Median (IQR) number of comorbidities		2 (1–3)	2 (1–3)	2 (1–3)	<0.0001
Median (IQR) CCI		5 (3–7)	5 (3–7)	5 (4–7)	<0.0001
Vaccine, n (%)		937 (81.8)	721 (89.1)	216 (64.3)	<0.0001
Doses, n (%)	0	208 (18.2)	88 (10.9)	120 (35.7)	<0.0001
	1	26 (2.3)	16 (2.0)	10 (3.0)	
	2	187 (16.3)	140 (17.39)	47 (14.09)	
	3	698 (61.0)	543 (67.1)	155 (46.1)	
	4	26 (2.3)	22 (2.7)	4 (1.2)	
Median (IQR) time since last vaccine dose		147 (84–204)	136 (82–191)	171.5 (98–227)	0.0001
**Symptoms**		954 (83.3)	639 (79.0)	315 (93.8)	<0.0001
Fever, n (%)		538 (47.0)	335 (41.4)	203 (60.4)	<0.0001
Cough, n (%)		410 (35.8)	257 (31.8)	153 (45.5)	<0.0001
Tachypnoea, n (%)		35 (3.1)	12 (1.5)	23 (6.9)	<0.0001
Ageusia, n (%)		17 (1.5)	13 (1.6)	4 (1.2)	0.79
Pharyngodynia, n (%)		162 (14.2)	132 (16.3)	30 (8.9)	0.001
Chills, n (%)		40 (3.5)	33 (4.1)	7 (2.1)	0.09
Asthenia, n (%)		410 (35.8)	290 (35.9)	120 (35.7)	1.00
Headache, n (%)		127 (11.1)	94 (11.6)	33 (9.8)	0.38
Myalgias, n (%)		184 (16.1)	134 (16.6)	50 (14.9)	0.48
Gastrointestinal symptoms, n (%)		171 (14.9)	120 (14.8)	51 (15.2)	0.88
Dyspnoea, n (%)		281 (24.5)	102 (12.6)	179 (53.3)	<0.0001
Nasal congestion, n (%)		53 (4.6)	49 (6.1)	4 (1.2)	<0.0001
Anosmia, n (%)		21 (1.8)	14 (1.7)	7 (2.1)	0.69
**Radiological findings**					
CT pneumonia, n (%)		449 (39.2)	193 (23.9)	256 (79.2)	<0.0001
GGO, n (%)		375 (32.8)	157 (19.4)	218 (64.9)	<0.0001
Consolidation, n (%)		227 (19.8)	79 (9.8)	148 (44.1)	<0.0001
Pulmonary Embolism, n (%)		21 (1.8)	12 (1.5)	9 (2.7)	0.17
**Therapy**					
Early treatment, n (%)		214 (65.6)	190 (66.7)	24 (58.5)	0.31
Monlupiravir, n (%)		242 (21.1)	224 (27.7)	18 (5.4)	<0.0001
Nirmatrelvir/ritonavir, n (%)		39 (3.4)	36 (4.5)	3 (0.9)	0.002
Remdesivir, n (%)		108 (9.4)	92 (11.4)	16 (4.8)	<0.0001
Casirivimab/Imdevimab, n (%)		110 (9.6)	91(11.3)	19 (5.7)	0.003
Sotrovimab, n (%)		130 (11.4)	118 (14.6)	12 (3.6)	<0.0001
Hospital infection, n (%)		304 (26.6)	233 (28.8)	71 (21.1)	0.007
Bacterial infection, n (%)		129 (11.3)	71 (8.8)	58 (17.3)	<0.0001

IQR: interquartile range; BMI: body mass index; COPD: chronic obstructive pulmonary disease; CCI: Charlson comorbidity index; CT: computer tomography; GGO: ground glass opacities.

**Table 2 viruses-15-00071-t002:** Characteristics of 389 patients with SARS-CoV-2 treated with antiviral therapies by treatment administered.

Variables		Molnupiravir (n = 242)	Nirmatrelvir/Ritonavir (n= 39)	Remdesivir (n = 108)	*p*-Value
Males, n (%)		135 (55.8)	22 (56.4)	60 (55.6)	1.00
Median (IQR) age, years		75 (64–83)	65 (53–79)	77 (68–84)	<0.0001^1^
Patient provenance	Emergency room	135 (55.8)	16 (41.0)	44 (40.7)	<0.0001
	Ward	78 (32.3)	14 (35.9)	64 (59.3)	
	Day-Hospital	29 (12.0)	9 (23.1)	0	
**Comorbidities**					
Immunodeficit, n (%)		42 (17.4)	13 (33.3)	11 (10.2)	0.004
Neurodevelopmental/neurodegenerative diseases, n (%)		54 (22.3)	5 (12.8)	41 (38.0)	0.002
Dementia, n (%)		24 (9.9)	1 (2.6)	20 (18.5)	0.01
Cerebrovascular events, n (%)		26 (10.7)	2 (5.1)	23 (21.3)	0.01
Cardiovascular diseases, n (%)		110 (45.5)	8 (20.5)	44 (40.7)	0.01
Heart failure, n (%)		370 (32.3)	241 (29.8)	129 (38.4)	0.005
Previous myocardial infarction, n (%)		147 (12.8)	101 (12.5)	46 (13.7)	0.58
Median (IQR) CCI		5 (4–7)	4 (3–6)	6 (4–7)	0.02^2^
**Other**					
Hospital infection, n (%)		67 (27.7)	11 (28.2)	59 (54.6)	<0.0001
Disease progression, n (%)		18 (7.4)	3 (7.7)	16 (14.8)	0.09

IQR: interquartile range; CCI: Charlson comorbidity index; PCT: procalcitonin; eGFR: estimated glomerular filtration rate; CRP: C-reactive protein. 1 Molnupiravir VS. Nirmatrelvir/ritonavir *p* = 0.001; Nirmatrelvir/ritonavir VS. Remdesivir *p* = 0.0004; 2 Molnupiravir VS. Nirmatrelvir/ritonavir *p* = 0.009; Nirmatrelvir/ritonavir VS. Remdesivir *p* = 0.01.

**Table 3 viruses-15-00071-t003:** Logistic regression analysis to assess the relationship between demographics, clinical characteristics and need to start oxygen therapy (n = 1145).

Variables		Univariate Analysis		Multivariate Analysis	
		OR (95% CI)	*p*-Value	OR (95% CI)	*p*-Value
Males		1.12 (0.86–1.44)	0.40	-	-
Age groups	<50 years	Ref.	Ref.	Ref.	Ref.
	50–59 years	2.05 (1.03–4.06)	0.04	2.87 (1.08–7.66)	0.04
	60–69 years	3.11 (1.71–5.67)	<0.0001	5.91 (2.47–14.1)	<0.0001
	70–79 years	3.57 (1.99–6.37)	<0.0001	7.27 (3.04–17.40)	<0.0001
	≥80 years	3.93 (2.25–6.88)	<0.0001	5.02 (2.12–11.87)	<0.0001
Weight, kg		1.00 (0.99–1.01)	0.30	-	-
**Comorbidity**					
BMI >30 kg/m^2^		1.28 (0.96–1.73)	0.10	-	-
Chronic renal failure		1.14 (0.82–1.61)	0.44	-	-
Dialysis		1.46 (0.63–3.36)	0.38	-	-
Immunodeficit		1.24 (0.87–1.78)	0.24	-	-
Transplanted		1.70 (0.64–4.50)	0.29	-	-
Rheumatological disease		1.16 (0.67–2.00)	0.61	-	-
Decompensated diabetes		1.81 (1.31–2.52)	<0.0001	1.37 (0.85–2.21)	0.20
Diabetes		1.16 (0.86–1.57)	0.34	-	-
Chronic liver disease		1.41 (0.84–23.36)	0.20	-	-
COPD/Emphysema		1.66 (1.22–2.26)	0.001	1.50 (0.93–2.41)	0.10
Haemoglobinopathies		0.60 (0.07–5.40)	0.65	-	-
Neurodevelopmental/neurodegenerative diseases		1.62 (1.23–2.13)	0.001	-	-
Dementia		2.07 (1.49–2.88)	<0.0001	2.06 (1.28–3.34)	0.003
Chromosopathies/hypoxia		2.43 (0.60–9.75)	0.21	-	-
Neuromuscular disease		0.77 (0.34–1.71)	0.52	-	-
Cerebrovascular events		1.30 (0.89–1.90)	0.18	-	-
Oncological disease		0.63 (0.43–0.93)	0.02	1.05 (0.59–1.86)	0.88
Metastasis		1.00 (0.56–1.78)	1.00	-	-
Terminal cancer		5.82 (2.22–15.27)	<0.0001	-	-
Haematological tumors		1.52 (0.92–2.50)	0.10	3.15 (1.54–6.44)	0.02
Solid tumors in chemotherapy		0.64 (0.28–1.49)	0.30	-	-
Haematological tumors in chemotherapy		1.47 (0.81–2.68)	0.21	-	-
Cardiovascular diseases		1.40 (1.08–1.81)	0.01	-	-
Heart failure		1.47 (1.13–1.92)	0.005	1.61 (1.07–2.43)	0.02
Previous acute myocardial infarction		1.11 (0.76–1.62	0.58	-	-
Hypertension		1.54 (1.19–1.99)	0.001	-	-
Number of comorbidities		1.22 (1.12–1.34)	<0.0001	-	-
CCI		1.11 (1.06–1.16)	<0.0001	-	-
Vaccine ≥ 2 doses		0.23 (0.17–0.32)	<0.0001	0.28 (0.18–0.42)	<0.0001
**Symptoms**		3.99 (2.49–6.40)	<0.0001	-	-
Fever		2.16 (1.67–2.80)	<0.0001	2.26 (1.56–3.29)	<0.0001
Cough		1.80 (1.39–2.33)	<0.0001	-	-
Tachypnoea		4.88 (2.40–9.93)	<0.0001	-	-
Ageusia		0.74 (0.24–2.28)	0.60	-	-
Pharyngodynia		0.50 (0.33–0.76)	0.001	-	-
Chills		0.50 (0.22–1.14)	0.10	-	-
Asthenia		0.99 (0.76–1.30)	0.97	-	-
Headache		0.83 (0.55–1.26)	0.38	-	-
Myalgias		0.88 (0.62–1.25)	0.48	-	-
Gastrointestinal symptoms		1.03 (0.72–1.47)	0.88	-	-
Dyspnoea		7.90 (5.86–10.65)	<0.0001	5.22 (3.44–7.90)	<0.0001
Nasal congestion		0.19 (0.07–0.52)	0.001	-	-
Anosmia		1.21 (0.48–3.02)	0.69	-	-
CT pneumonia		10.21 (7.58–13.77)	<0.0001		
GGO		7.67 (5.78–10.19)	<0.0001	3.98 (2.69–5.89)	<0.0001
Consolidation		7.27 (5.30–9.98)	<0.0001	2.93 (1.87–4.60)	<0.0001
Pulmonary Embolism		1.83 (0.76–4.38)	0.18	-	-
Early treatment		0.71 (0.36–1.38)	0.31	-	-
Antiviral		0.16 (0.11–0.23)	<0.0001	-	-
Molnupiravir		0.15 (0.09–0.24)	<0.0001	0.11 (0.06–0.20)	<0.0001
Nirmatrelvir/ritonavir		0.19 (0.06–0.63)	0.007	0.16 (0.04–0.67)	0.01
Remdesivir		0.39 (0.23–0.67)	0.001	0.22 (0.11–0.43)	<0.0001
Casirivimab/Imdevimab		0.47 (0.28–0.79)	0.004	0.34 (0.18–0.66)	0.002
Sotrovimab		0.22 (0.12–0.40)	<0.0001	0.21 (0.10–0.45)	<0.0001
Hospital infection		0.66 (0.49–0.90)	0.008	1.29 (0.82–2.04)	0.27
Bacterial infection		2.17 (1.49–3.15)	<0.0001	2.11 (1.25–3.59)	0.006

## Data Availability

The data that support the findings of this study are available from the corresponding author, upon reasonable request.

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
