# Peer review of "Impact of Early SARS-CoV-2 Antiviral Therapy on Disease Progression"

_viruses, 2022, doi:10.3390/v15010071_

Round 1
Reviewer 1 Report
Many typos are present in the text (e.g., line 46 SARS-Cov-2). Please re-read it and fix them.
Line 70: what do you mean by admission?
In line 75, you wrote, “Criteria for their prescription are/were:” I suggest modifying it with “. Criteria for their prescription were”.
The number of people treated with monoclonal antibodies is 237, while the sum of sotrovimab and casirivimab/imdevimab is 240. Therefore, please recheck your data to fix it.
I think you should add some sentences about the setting of your patients. It is important to understand why the percentage of people with severe disease is high compared to other studies.
I think you should add this study to your discussion https://doi.org/10.1093/jac/dkac256.
Author Response
Dear Editor,
We want to thank you for allowing us to revise our manuscript and thank the reviewers for their careful reading and thoughtful comments on the previous draft.
We have carefully considered their comments in preparing our revision, which has resulted in a more precise, compelling, and broader paper.
Please, find below our point-by-point response to reviewers (all revisions have been written in red):
Reviewer (R): Many typos are present in the text (e.g., line 46 SARS-Cov-2). Please re-read it and fix them.
AR: Thank you for having read our paper carefully. We re-read the article and fixed all these issues.
R: Line 70: what do you mean by admission?
AR: We modified this sentence to We collected information on medical history, vaccination status, computer tomography 69 (CT) findings, symptoms’ at the admission, and therapies.
R: In line 75, you wrote, “Criteria for their prescription are/were:” I suggest modifying it with “. Criteria for their prescription were”.
AR: Thank you for the suggestion. We modified the sentence.
R: The number of people treated with monoclonal antibodies is 237, while the sum of sotrovimab and casirivimab/imdevimab is 240. Therefore, please recheck your data to fix it.
AR: Thank you again for your attention. We rechecked the data and confirmed that 240 is the correct number. We apologise for this typo.
R: I think you should add some sentences about the setting of your patients. It is important to understand why the percentage of people with severe disease is high compared to other studies.
AR: We added a sentence about our setting as requested.
R: I think you should add this study to your discussion https://doi.org/10.1093/jac/dkac256.
AR: Thank you for your suggestion. We added a sentence about this interesting paper.
Reviewer 2 Report
In this retrospective cohort study, the authors showed that the vaccination and the early antiviral or the monoclonal antibodies treatment was associated with the reduced COVID-19 disease progression.
This is an important exhaustive study with the interesting results, in light of managing and preventing the infections. This study is well designed and the manuscript is well written. The authors need to spell out acronyms/abbreviations (AIFA, DDI, ARDS, P/F) in the manuscript. Once the authors correct these, this manuscript can be accepted for publication.
Author Response
Dear Editor,
We want to thank you for allowing us to revise our manuscript and thank the reviewers for their careful reading and thoughtful comments on the previous draft.
We have carefully considered their comments in preparing our revision, which has resulted in a more precise, compelling, and broader paper.
Please, find below our point-by-point response to reviewers (all revisions have been written in red):
Reviewer 2
In this retrospective cohort study, the authors showed that the vaccination and the early antiviral or the monoclonal antibodies treatment was associated with the reduced COVID-19 disease progression.
This is an important exhaustive study with the interesting results, in light of managing and preventing the infections. This study is well designed and the manuscript is well written. The authors need to spell out acronyms/abbreviations (AIFA, DDI, ARDS, P/F) in the manuscript. Once the authors correct these, this manuscript can be accepted for publication.
AR: Thank you for having read our manuscript. We spell out the abbreviations as requested.